# Analysis of Accelerometer Data Using Random Forest Models to Classify the Behavior of a Wild Nocturnal Primate: Javan Slow Loris (*Nycticebus javanicus*)

Amanda Hathaway [1], Marco Campera [1,2,3], Katherine Hedger [2], Marianna Chimienti [4], Esther Adinda [2], Nabil Ahmad [2], Muhammed Ali Imron [5] and K. A. I. Nekaris [1,2,*]

[1] School of Social Sciences, Oxford Brookes University, Oxford OX3 0BP, UK; ahathaway2014@gmail.com (A.H.); mcampera@brookes.ac.uk (M.C.)

[2] Little Fireface Project, Cisurupan, Bandung 40131, Indonesia; research@littlefireface.org (K.H.); estheradinda810@gmail.com (E.A.); ahmadnabilff@gmail.com (N.A.)

[3] Department of Biological and Medical Sciences, Oxford Brookes University, Oxford OX3 0BP, UK

[4] Centre d'Etudes Biologiques de Chizé, 405 Route de Prissé la Charrière, 79360 Chizé, France; mariannachimienti@gmail.com

[5] Faculty of Forestry, Universitas Gadjah Mada, Yogyakarta 55281, Indonesia; maimron@ugm.ac.id

* Correspondence: anekaris@brookes.ac.uk

**Abstract:** Accelerometers are powerful tools for behavioral ecologists studying wild animals, particularly species that are difficult to observe due to their cryptic nature or dense or difficult to access habitats. Using a supervised approach, e.g., by observing in detail with a detailed ethogram the behavior of an individual wearing an accelerometer, to train a machine learning algorithm and the accelerometer data of one individual from a wild population of Javan slow lorises (*Nycticebus javanicus*), we applied a Random Forest model (RFM) to classify specific behaviors and posture or movement modifiers automatically. We predicted RFM would identify simple behaviors such as resting with the greatest accuracy while more complex behaviors such as feeding and locomotion would be identified with lower accuracy. Indeed, resting behaviors were identified with a mean accuracy of 99.16% while feeding behaviors were identified with a mean accuracy of 94.88% and locomotor behaviors with 85.54%. The model identified a total of 21 distinct combinations of six behaviors and 18 postural or movement modifiers in this dataset showing that RFMs are effective as a supervised approach to classifying accelerometer data. The methods used in this study can serve as guidelines for future research for slow lorises and other ecologically similar wild mammals. These results are encouraging and have important implications for understanding wildlife responses and resistance to global climate change, anthropogenic environmental modification and destruction, and other pressures.

**Keywords:** animal behavior; supervised machine learning; random forest model

## 1. Introduction

Understanding behavior and physiology of animals in their natural environments is fundamental to ecology [1]. For centuries, animal behaviorists and ecologists have relied on direct observations to gather insights on animals' activities. Wild animals may be difficult to observe [2,3]; direct observations can introduce observer bias [1,4] as well as the potential to affect animal behavior [3,5–7]. The use of bio-loggers, animal-borne devices that provide data on animal movement, behavior, and physiology without the need for direct observation, have proven to be powerful tools to study animal behavior. GPS trackers, video cameras, temperature loggers, depth recorders, physiological loggers, etc. have aided behavioral ecology researchers to observe and understand animal behavior [8,9]. Animal-borne accelerometers, devices that provide data of static and dynamic acceleration, are

particularly powerful tools that aid in the study of animal behavior and have applications in the fields of captive animal welfare [10–13], behavioral ecology [14–16], and evolutionary studies [16–18].

Accelerometers have enabled animal behavior researchers to study species that may otherwise be very difficult or impossible to observe directly, either due to their cryptic nature, behaviors or qualities that make them less easily detected by predators or prey [19], or the difficulty of accessing or navigating their environments. The first study to utilize accelerometers to discern behavioral patterns was conducted by Yoda et al. (1999) [20] to classify the movement behaviors of Adélie penguins (*Pygoscelis adeliae*). Since then, improvements in technology (e.g., logger size as well as battery and storage capacity) enabled opportunities to research a wider range of species that were previously inaccessible. For example, Nakamura, Goto, and Sato (2015) [21] attached accelerometers to sun fish that dive up to 200 m for extended periods of time making them nearly impossible to observe directly. Nocturnal mammals are particularly difficult to observe, and researchers often rely on other metrics such as vocalizations to determine abundance or radio tracking [22]. Although radio tracking is useful to discern general movement patterns and social organization, animals may remain completely out of view. To date, accelerometers have been used most widely in studies of birds and marine mammals [15]. Only a few studies have used accelerometers to study primate behavior [13,15,23–26] and the majority of these sought to identify broad activity categories rather than specific behaviors. Even fewer studies have focused on nocturnal primates (e.g., those for which accelerometers would be the most useful).

Another benefit of bio-logging is the elimination of observer bias, since the presence of humans can unintentionally influence animal behavior [6,7,27]. Even when animals are habituated, human presence can affect the behavior of non-habituated animals and influence their interactions with the habituated focal animals [5,28]. Direct observations are also limited by the boundaries of our own physical and sensory abilities; our individual experiences implicitly cause us to focus on certain events and subjects more than others [1,2,4,29].

Modern accelerometers tend to last for longer periods than older models and collect data continuously for an animal's entire active period, which a human observer is rarely capable of unless through video recording. Despite improvements, battery life of accelerometers continues to be a major challenge. For instance, battery life can be affected by weather and humidity; seasonal variation must also be considered when planning deployment and retrieval of devices in the field [30–32]. Battery life of accelerometers is also affected by the frequency interval at which the accelerometer is set to record data [32]. High recording frequencies (>25 Hz) drain the device's battery more quickly than low frequencies. Some research has been carried out to determine whether lowering recording frequency to extend battery life significantly affected precision of behavior classification. Hounslow et al. (2019) [33] tested a range of frequencies (1–30 Hz) on lemon sharks (*Negaprion brevirostris*) and found that classification precision of fine-scale behaviors did not decrease significantly until recording frequency reached as low as 5 Hz. McGowan et al. (2022) [34] compared two accelerometer models and found that the model with higher capacity and higher recording frequency outperformed the other. Generally, it is recommended to use mid to high range frequencies when attempting to classify more complex behaviors, but low frequencies are acceptable to classify less complex behaviors and will extend the life of the device's battery.

The detailed three-dimensional datasets derived from accelerometers can be used to identify specific animal behaviors and require complex stochastic analytical methods to infer behavior [35]. Additionally, the raw accelerometer dataset only provides acceleration and orientation information so various models can be used to infer the actual behaviors. Machine-learning models are used to develop an algorithm that automatically identifies patterns within the dataset. Broadly, there are two categories of machine-learning algorithms: supervised and unsupervised [36]. The most important difference between supervised and unsupervised learning algorithms is their inputs and outputs. Supervised

learning algorithms produce classifications based on the labels researchers assign to the training dataset while unsupervised learning algorithms produce associative clusters of data using pattern recognition. There are several ways to cluster the data, which means there are multiple possible outcomes, so researchers must indicate a similarity measure for the model to follow. Unsupervised learning algorithms are more complex and less precise than supervised algorithms but may be used to identify the labels that can then be applied to a supervised learning algorithm [37].

A supervised learning algorithm is used by behavioral ecologists when an ethogram, a list of distinct behaviors and their descriptions, is already known [38–40]. These behaviors are used to label a portion of the training dataset. A statistical model is then applied to the data subset to classify behaviors using acceleration signatures [15,41]. This method requires that the researchers have pre-existing knowledge of the species, which is often not the case with many cryptic or difficult to access species that researchers know very little about. In these cases, an unsupervised learning algorithm is used which forgoes the need for direct observations [15,42,43]. Several supervised learning models have been used to develop classification algorithms for animal acceleration data including decision trees, support vector machines, and random forest models [44,45].

One group of primates that lend themselves particularly well to wearing accelerometers are the Lorisidae—African pottos (Perodicticinae) and Asian lorises (Lorisinae) Their cryptic lifestyles make them particularly difficult to observe, but at the same time, their non-jumping movements that are often slow can be picked up well by an accelerometer [46]. Direct observations by human researchers have provided detail about behaviour of slow lorises in particular in the wild, but for significant portions of time animals are out of view [47]. Despite the challenge of following these nocturnal primates, they have been shown to eat gum, nectar, and insects; their activity patterns are influenced by weather and moon phase; they go into torpor often in dense foliage where this behavior may be missed; and are frequently social, a behavior said to be rare for nocturnal primates [47–50]. Although all slow loris species are arboreal and prefer tree connectivity, several slow loris species occur in agroforests with reduced canopy connectivity that may disrupt loris activities or impact their energetics [51–54]. Understanding the impacts of these factors is particularly important for Javan slow lorises, which are classified as Critically Endangered by the International Union of the Conservation of Nature (IUCN) Red List due to intense deforestation and fragmentation for agriculture [52,55]. Indeed, as natural forests shift more and more to agriculture, there is a call to understand behavior and ecology within agroforestry matrix environments [56,57].

Here we present a case study of applying a supervised learning approach to train a model to identify behaviors from accelerometer data of a wild Javan slow loris (*Nycticebus javanicus*), from a well-known population occurring within an agroforest in Indonesia. Using direct behavioral observation data, we applied a supervised learning approach to train a random forest model [58,59]. Next, we validated the accuracy of the model's predictions against our observations and present the results. It is predicted that movement complexity will affect the model's classification accuracy. We predicted resting behaviors would be classified with highest accuracy and feeding and locomotor behaviors such as climbing and walking would be classified with lower accuracy. We divided the results by broad behavioral categories: Locomotive, Feeding, and Resting. This is the first time accelerometry and machine learning have been applied to wild slow lorises to identify specific behaviors. For this reason, we tested the method with a single animal as proof of principle. The results imply exciting applications of accelerometry to behavioral ecology of cryptic arboreal mammals.

## 2. Materials and Methods

Using data extracted from an accelerometer worn by a wild male Javan slow loris, we developed an algorithm using a random forest model to identify behaviors. Direct behavioral observations were used to validate the algorithm. The study area lies outside

the village of Cipaganti in West Java, Indonesia (7°16′44.30″ S, 107°46′7.80″ E, 1200 m asl) [52] and is part of the Little Fireface Project (LFP), which has been consistently studying a wild population of Javan slow lorises since 2011. LFP is the longest continuous research project of any nocturnal primate species, which is ideal to validate the methods of this study. Cipaganti is located on the Gunung Puntang Mountain at 1345 m above sea level [60] and exists nearby, but outside a strictly protected nature reserve, Gunung Papandayan. The landscape of Cipaganti is agroforest, which is characterized by patchworks of forest fragments, agricultural fields, and human settlements [61] (Figure 1). The climate of the region is tropical rainforest with annual precipitation exceeding 2500 mm [52] and temperatures remain relatively constant throughout the year but vary more between day and night [62]. Between January and August 2022, minimum lows reached 22 °C at night while maximum highs reached 35 °C during the day [60].

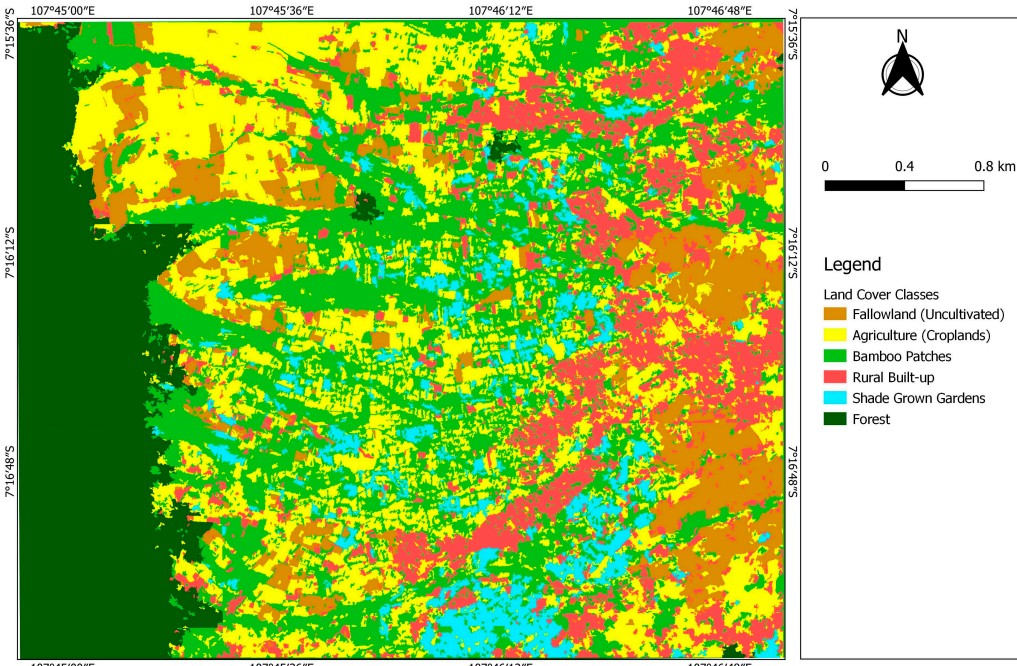

**Figure 1.** A map of the study area of Cipaganti, Java, Indonesia, showing the anthropogenic landscape in which the Javan slow lorises live.

## 2.1. Field Methods

A team of three to five people from LFP, using protocol recommended by Nekaris, Munds, and Pimley (2020) [63], captured an adult male loris on 7 March 2022 and fitted him with a collar affixed with a radio transmitter and accelerometer and recaptured him on 11 April 2022 to retrieve the accelerometer. Any medical check-ups, sample collection, measurements, notes, and collar fittings are conducted in situ and without the use of anesthetic (Figure 2).

We recorded the target slow loris' behavior between the hours of 18:30 and 23:00 on 10, 15, and 18 March 2022. We recorded general behavior and positional and locomotor behaviors using a scan sampling method at five-minute intervals plus ad libitum observations. For the purposes of this study, and based on validation of an accelerometer in captivity, we used a reduced ethogram combining six behavioral categories alongside 11 postures (Tables 1–3). Since the captive slow loris was on her own, we did not have validation data to include any social behaviors for the current study.

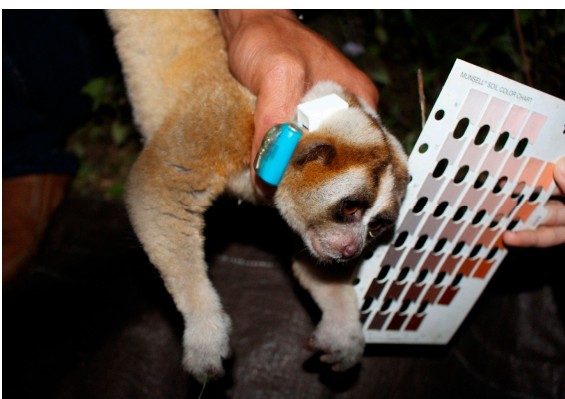

**Figure 2.** Javan slow loris (*Nycticebus javanicus*) wearing a 17 g VHF radio collar (blue cell attached with shrink wrap covered cable tie) with affixed 2.5 g accelerometer (white box). Photo courtesy of Little Fireface Project.

**Table 1.** Ethogram of *Nycticebus javanicus* behaviors and behavioral modifiers used in this study to label the accelerometer training dataset used to validate the Random Forest model.

| Behavior | Abbreviation | Description |
|----------|--------------|-------------|
| Alert | al | Remain stationary as in "rest" but active observation of environment or observer |
| Explore | ex | Meandering movement associated with looking for food or exploring the habitat |
| Feeding | fe | Consumption of a food item |
| Travel | tr | Continuous, directed movement from one location to another |
| Groom | gr | Autogroom, lick, or use tooth comb on own fur |
| Rest | re | Remain stationary, often with body hunched, eyes open |

**Table 2.** Ethogram of *Nycticebus javanicus* positional behavior used in this study to label the accelerometer training dataset used to validate the Random Forest model.

| Posture | Abbreviation | Description |
|---------|--------------|-------------|
| Sit | si | Remain stationary with body hunched and head erect |
| Stand | st | Remain stationary supported on all fours, limbs extended |
| Horizontal suspension 1 | H1 | Hanging from one foot |
| Horizontal suspension 2 | H2 | Hanging from two feet or bipedal standing |
| Horizontal suspension 3 | H3 | Hanging from three feet |
| Horizontal suspension 4 | H4 | Hanging from four feet |
| Vertical suspension 2 | V2 | Hanging towards the side of a support with 2 feet (e.g., when foraging or observing) |
| Vertical suspension 3 (up or down) | V3u V3d | Hanging towards the side of a support with 3 feet, either facing upwards or downwards |
| Vertical suspension 4 (up or down) | V4u V4d | Hanging towards the side of a support with 4 feet, either facing upwards or downwards |

**Table 3.** Ethogram of *Nycticebus javanicus* locomotor behavior used in this study to label the accelerometer training dataset used to validate the Random Forest model.

| Locomotion | Abbreviation | Description |
|---|---|---|
| Walk | wa | Quadrupedal walking on 0° to 45° support |
| Racewalk | rw | Fast quadrupedal walking on 0° to 45° support |
| Suspensory walk | sw | Locomoting while hanging on 0° to −45° support |
| Bridge | bg | Climbing from one support to the next (trunk or branches of same or different trees) stretching over a gap of more than 15 cm |
| Climb up | cu | Moving upwards on +/−45° to +/−90° support |
| Climb down | cd | Moving downwards on +/−45° to +/−90° support |
| Climb horizontally | ch | Moving horizontally through 0° to +/−45° support |

## 2.2. Materials

We used a Technosmart Axy 5s accelerometer, with dimensions of 22 mm × 13 mm × 10 mm, weighing 2.5 g, mounted to a Lotek VHF radio collar using two zip ties (Figure 2). The combined weight of the collar and accelerometer is around 19 g, which is below the recommended 5% of the animal's body mass [64,65]. We set the accelerometers to record at an interval of 25 Hz. The device in this study lasted the manufacturer suggested 60 days at this rate [32].

## 2.3. Data Analysis

We conducted all data processing in Microsoft Excel and we used R and RStudio version 2022.02.2+485 to run the Random Forest model and validate the R script. Raw data from the Technosmart Axy 5s accelerometer provided information on acceleration and orientation through measurements of 15 variables (Table 4).

**Table 4.** List of variables in raw accelerometer data from Technosmart Axy 5s model accelerometer.

| Accelerometer Variables | | |
|---|---|---|
| AccX | AccY | AccZ |
| Static_DorsoVentral | Static_Lateral | Static_BackForward |
| Amplitude_DorsoVentral | Amplitude_Lateral | Amplitude_BackForward |
| Dynamic_DorsoVentral | Dynamic_Lateral | Dynamic_BackfForward |
| Pitch | ODBA_vec | Amplitude_Pitch |

We extracted accelerometer data between 10–18 March 2022, which corresponded to the direct observations taken in the field. Aligning the timestamps from both datasets, we added labels to the raw dataset with behaviors from the direct behavioral observations. Behaviors are recorded the moment the stopwatch signals and the accelerometer records data 25 times per second. We thus labelled all 25 datapoints with the same behavior. For instance, if a behavior is recorded for the time stamp 18:20 and there are 25 datapoints corresponding to the time 18:20:00, all 25 accelerometer datapoints are labelled with the same behavior. The labelled subset of accelerometer data consists of a total of 2900 datapoints. We divided the data into three parts and subsets (locomotive—explore and travel; feeding behaviors—feeding on gum, nectar, insects, etc.; and resting behaviors—alert, groom, rest) based on broad behavioral categories, then we ran the Random Forest model three times, once for each subset.

We used the labelled accelerometer dataset to train a Random Forest model to classify behaviors. We ran a Random Forest script in RStudio derived from the one used in Nekaris et al. (2022) [13]. See Appendix A for the Random Forest script we used.

Random Forest models can be defined as:

"a classifier consisting of a collection of tree-structured classifiers {h(x,k), k = 1,...} where the {k} are independent identically distributed random vectors and each tree casts a unit vote for the most popular class at input x" [58]

The benefits to using random forests as opposed to a single decision tree are an increase in prediction accuracy and outputs of variable importance and prediction uncertainty [58,59]. A single decision tree is prone to overestimating the importance of certain variables and overfitting classifications. Random forests avoid this problem by introducing two random selection processes each time a tree is grown so that each tree is different from the next, thus increasing variability. Variability reduces the risk of overfitting and overemphasis of the importance of certain variables. Once all of the trees in the forest have made their predictions, the predictions are aggregated, with the most popular being the result of the model. The nodes of a decision tree terminate when the data included in each node cannot be classified any further, thus they are 'pure'. The purity or impurity of each node is quantified with the Gini impurity index formula. The Gini index tends towards zero when the subset is pure or contains only one kind of class (in this case, behaviors). The model runs a subset through a decision tree which splits the data at nodes with the goal of minimizing the Gini impurity index.

$$G = \sum_{i=1}^{n} p_i(1 - p_i)$$

where $n$ is the number of behavioral classes and $p_i$ is the proportion of each class in a set of observations.

First, a training subset was randomly selected from the labelled dataset while the remaining 30% is used as a validation dataset, which was then used to test the accuracy of the Random Forest model predictions. Once we built the model, we used it to predict the behaviors of the validation dataset. We then compared the predicted behaviors to the observed behaviors and produced a confusion matrix to assess the accuracy of the model.

## 3. Results

The Random Forest model identified 21 separate modified behaviors, wherein the raw accelerometer variables yielded a mean overall prediction accuracy of 91.6% for the training dataset and 94.6% for the validation dataset across all three behavior categories. The behavior identified with the lowest accuracy in the training dataset was tr_wa (travel_walk) at 74.08% and the behavior identified with the lowest accuracy in the validation dataset was ex_bg (explore_bridge) at 80%. Resting behaviors were identified with the highest accuracy—99.16% from the resting training dataset. Locomotive behaviors were identified with the least accuracy—85.54% from locomotive training dataset. The sections that follow are the results of the Random Forest model validation presented by broad behavioral categories, locomotive, feeding, and resting.

### 3.1. Locomotive Behaviors

Locomotive behaviors were identified with a mean accuracy of 85.54%. Explore_climb down (ex_cd) had the highest prediction accuracy (94.4%). Travel_walk (tr_wa) had the lowest prediction accuracy (74.08) and was confused most often with explore_climb down (ex_cd). Static_DorsoVentral was the most important variable to predict locomotive behaviors including travel and explore (Table 5), based on mean decrease accuracy and decrease GINI (Figure 3). Static_Lateral was the most important classifier (Figure 4).

**Table 5.** Results of the Random Forest classification to assess the predictive power of the variables retrieved from a three-axis accelerometer in assessing the locomotive behaviors of a wild Javan slow loris. Prediction accuracy, main confusing behaviors, and the importance of variables in the Random Forest classifier were based on the training set of data.

| Behavior | Prediction Accuracy (%) | Main Confusing Behavior(s) (% Error) | Most Important Variables in Random Forest Classifier | | |
|---|---|---|---|---|---|
| | | | 1st Variable | 2nd Variable | 3rd Variable |
| ex_bg | 85.72 | ex_cd (11.9) | Static_Lateral | Static_DorsoVentral | accY |
| ex_cd | 94.4 | tr_cu; tr_wa (1.3) | Static_DorsoVentral | Static_Lateral | accZ |
| ex_ch | 77.42 | tr_cd (12.9) | Static_DorsoVentral | Static_Lateral | accY |
| ex_cu | 94.32 | tr_bg (1.9) | Static_DorsoVentral | Static_Lateral | accZ |
| ex_wa | 81.63 | ex_cd; ex_cu (9.18) | Static_DorsoVentral | accZ | accY |
| tr_bg | 82.44 | ex_cd (9.16) | Static_Lateral | Static_DorsoVentral | Pitch |
| tr_cd | 86.86 | ex_cd; tr_cu (2.85) | Static_Lateral | Static_DorsoVentral | Pitch |
| tr_cu | 94.12 | ex_cu; tr_bg (1.96) | Static_DorsoVentral | Static_Lateral | accZ |
| tr_rw | 83.34 | ex_cd (16.6) | Static_DorsoVentral | Static_Lateral | Pitch |
| tr_sw | 86.57 | ex_cu (8.95) | Static_Lateral | Static_DorsoVentral | Pitch |
| tr_wa | 74.08 | ex_cd (19.75) | Static_DorsoVentral | Static_Lateral | Pitch |

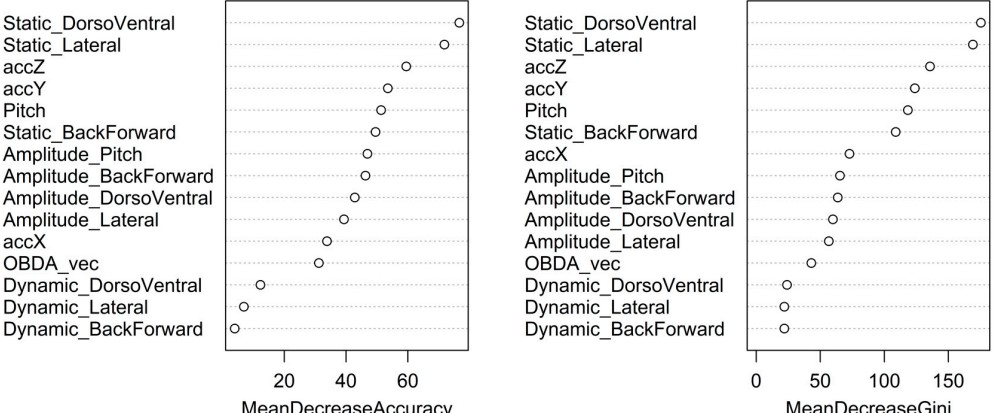

**Figure 3.** Mean decrease accuracy and mean decrease GINI of the predictor variables for locomotive behaviors included in the Random Forest classifier ordered by importance to the model.

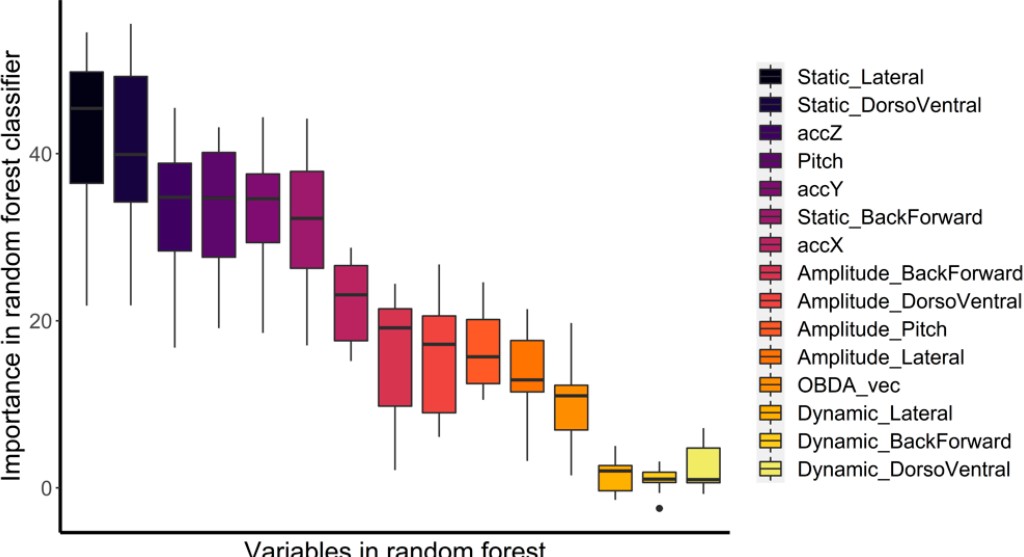

**Figure 4.** Box plot of variable importance as a classifier for locomotive behaviors included in the Random Forest model. Points represent outliers.

### 3.2. Feeding Behaviors

Feeding behaviors were classified with a mean accuracy of 94.88%. Feeding_horizontal 3 (fe_h3) had the highest prediction accuracy (100%). Feeding_vertical 2 (fe_v2) had the lowest prediction accuracy (93.75%) and was confused most often with feeding_vertical 4 (fe_v4) (Table 6). AccZ was the most important classifier (Figures 5 and 6). Acceleration on the Z axis (accZ) was the most important variable to predict feeding behaviors (see Table 6), based on mean decrease accuracy and decrease GINI (Figure 6).

**Table 6.** Results of the Random Forest classification to assess the predictive power of the variables retrieved from a three-axis accelerometer in assessing the feeding behaviors of a wild Javan slow loris. Prediction accuracy, main confusing behaviors, and the importance of variables in the Random Forest classifier were based on the training set of data.

| Behavior | Prediction Accuracy (%) | Main Confusing Behavior (% Error) | Most Important Variables in Random Forest Classifier | | |
|---|---|---|---|---|---|
| | | | 1st Variable | 2nd Variable | 3rd Variable |
| Fe_h3 | 100 | Na | accZ | Static_DorsoVentral | accX |
| Fe_h4 | 90.9 | Fe_v2 (9.09) | accX | Static_BackForward | Pitch |
| Fe_v2 | 93.75 | Fe_h4 (6.25) | accZ | accX | Static_DorsoVentral |

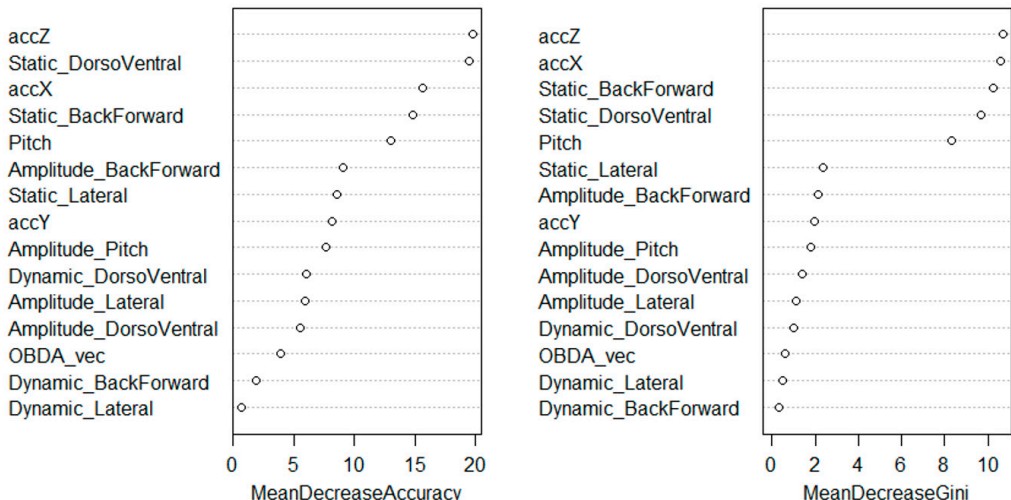

**Figure 5.** Mean decrease accuracy and mean decrease GINI of the predictor variables for feeding behaviors included in the Random Forest classifier ordered by importance to the model.

### 3.3. Resting Behaviors

Resting behaviors were identified with a mean accuracy of 99.16%. Alert_horizontal 2 (al_h2) had the highest prediction accuracy (100%). Alert_horizontal 4 (al_h4) had the lowest prediction accuracy (94.12%) and was confused most often with alert_vertical 4 down (al_v4d) (Table 7). Static_DorsoVentral was the most important variable to predict resting behaviors (see Table 7), based on mean decrease accuracy and decrease GINI (Figure 7). Static_DorsoVentral was the most important classifier (Figure 8).

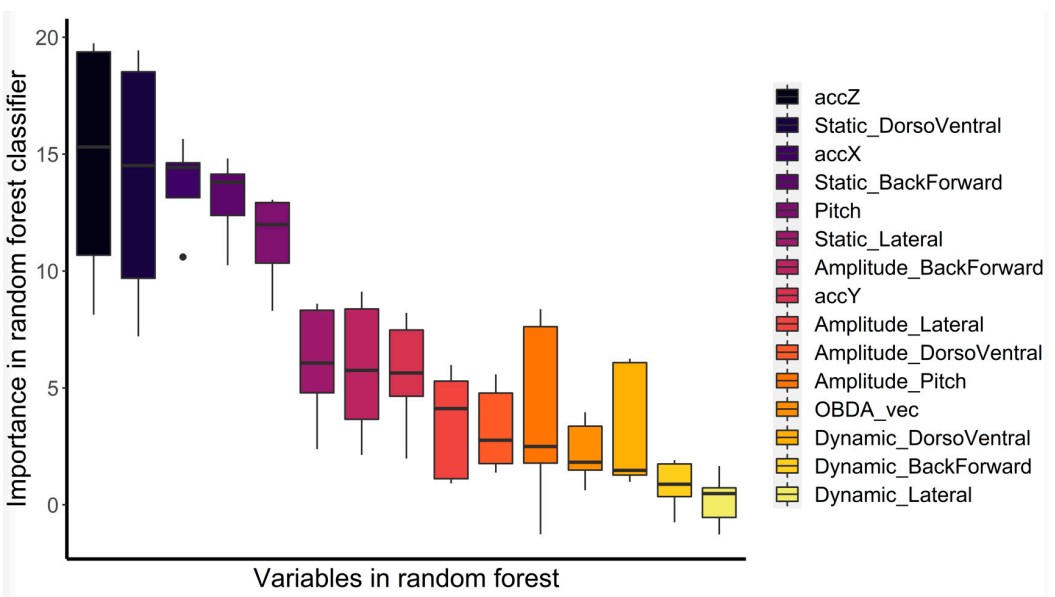

**Figure 6.** Box plot of variable importance as a classifier for feeding behaviors included in the Random Forest model. Points represent outliers.

**Table 7.** Results of the Random Forest classification to assess the predictive power of the variables retrieved from a three-axis accelerometer in assessing the resting behaviors of a wild Javan slow loris. Prediction accuracy, main confusing behaviors, and the importance of variables in the Random Forest classifier were based on the training set of data.

| Behavior | Prediction Accuracy (%) | Main Confusing Behavior(s) (% Error) | Most Important Variables in Random Forest Classifier | | |
|---|---|---|---|---|---|
| | | | **1st Variable** | **2nd Variable** | **3rd Variable** |
| Al_h2 | 100 | Na | accZ | Static_DorsoVentral | Static_Lateral |
| Al_h4 | 94.12 | Al_V4d (5.88) | Static_DorsoVentral | accZ | Amplitude_Lateral |
| Al_si | 100 | Na | Static_DorsoVentral | accZ | Static_Lateral |
| Al_st | 100 | Na | Static_DorsoVentral | accZ | Static_Lateral |
| Al_v4d | 100 | Na | Static_DorsoVentral | accZ | Static_BackForward |
| Gr_si | 100 | Na | Static_DorsoVentral | Static_Lateral | accZ |
| Re_sb | 100 | Na | Static_Lateral | Static_DorsoVentral | accY |

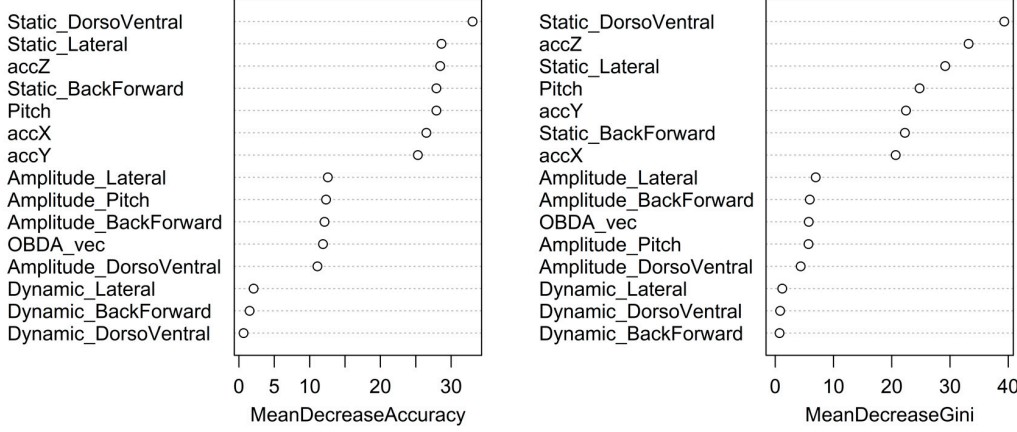

**Figure 7.** Mean decrease accuracy and mean decrease GINI of the predictor variables for resting behaviors included in the Random Forest classifier ordered by importance to the model.

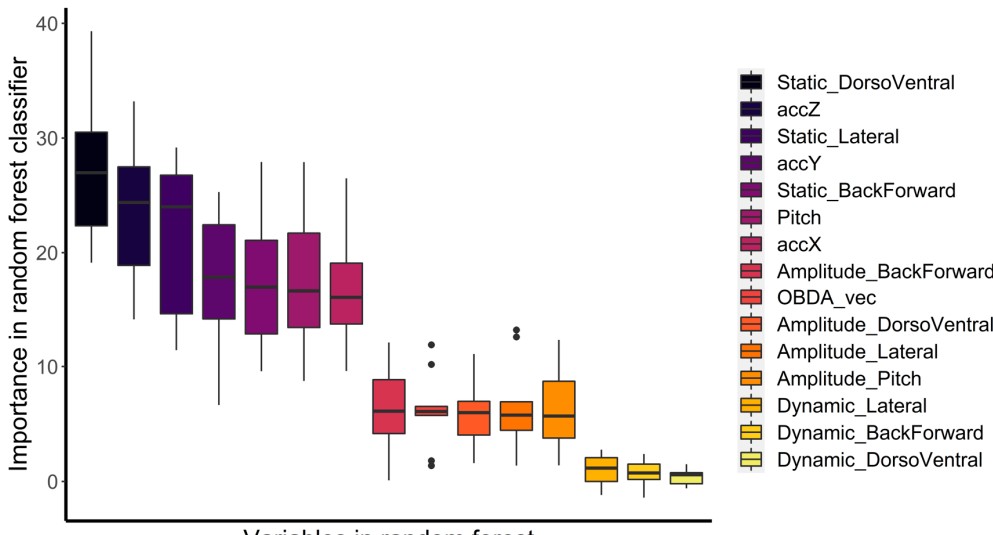

**Figure 8.** Box plot of variable importance as a classifier for resting behaviors included in the Random Forest model. Points represent outliers.

## 4. Discussion

Our aim in this study was to see if an accelerometer could accurately predict behaviors of a wild Javan slow loris, compared to those recorded by a human observer. By combining direct behavioral observation data and accelerometer data within a Random Forest model framework, we have successfully identified 21 combinations of six behaviors and 18 movement/position modifiers from a wild Javan slow loris with a mean accuracy of 91.6% in the training datasets and 94.6% in the validation datasets. The Random Forest model identified resting behaviors with the greatest accuracy (99.16%) and locomotive behaviors with the lowest accuracy (85.54%), which is consistent with the results of similar studies in other species [13,15]. The reason for this disparity may be due to fundamental differences between the two behavioral categories. Locomotive behaviors are more complex and varied than resting behaviors [65], which likely increases the chances of confusion in the Random Forest model. Evidence of this complexity can be seen by looking at the number of combinations of behavior and position or locomotion; these included 11 combinations for locomotive behaviors (see Table 5) and seven combinations for resting behaviors (see Table 7). Interestingly, only three combinations were identified for feeding behaviors (see Table 6), yet these were not identified with as high accuracy as resting behaviors. This may be due to the relatively small sample size of feeding behaviors (150 datapoints) as compared with resting (400 datapoints) or locomotive (2125 datapoints). Further study is needed to determine how sample size influences classification accuracy.

Within locomotive behaviors (explore and travel), we found seven positional or locomotive modifiers. The primary difference between the two behaviors, explore and travel, is the perceived intentionality of the loris by the human observer. The ethogram (see Table 2) defines explore as "movement associated with looking for food or exploring the habitat". This implies that the purpose of the loris' movement is to search for food, or simply exploring their environment. Travel is defined as "continuous, directed movement from one location to another", which implies the purpose of the movement is to simply get to another location. Both behaviors involve travelling from one place to another, and so, to an accelerometer, look very similar and the device may not be sensitive enough to discern visual and olfactory searching. Human observers are still important to interpret subtle differences in behaviors such as these. For the purposes of future accelerometer studies, it may be beneficial to classify explore and travel behaviors under one behavioral category to avoid confusion until improvements in accelerometer technology make them sensitive enough to discern the nuances of behavior.

Across all behavior categories the most important variables were static_DorsoVentral, static_Lateral, accZ, and pitch. This somewhat reflects the results of Nekaris et al. (2022) [13], which examined accelerometer data from a captive individual of a different species of loris, *Nycticebus bengalensis*. They found Static_Lateral and Static_DorsoVentral to be the first and second most important variables respectively to predict behaviors while accZ was the third most important variable for just one behavior. In the current study, the second and third most important variables in the feeding category were undetermined due to the fact that across the three distinct feeding behaviors, there were three distinct variables in second and third place. Between all three behavior categories, feeding behaviors had the smallest sample size. Replication of the model with a much larger sample size might reveal truer variable importance for feeding behaviors.

A greater sample size is needed to further validate and increase the reliability of the algorithm before we can run unlabeled data with confidence. The objective is to have ample sample size with the maximum number of subjects and behavior variability to build an algorithm that is robust enough to apply to any Javan slow loris accelerometer data without the need for correlating behavioral data. A reliable algorithm may potentially be further tested and then applied to any species with similar morphology and ecology [66] such as other loris species although some scientists caution the use of one algorithm across different species [67]. This study provides proof of method that can be applied to any lorisid species with an established ethogram.

Comparison of results with previous studies is difficult to do for a variety of reasons. Of the few studies that seek to investigate primate behavior using accelerometers, only two [13,15] classify behaviors using similar methods and present their results using the same metrics as those in the present study i.e., model accuracy and variable importance. Table 8 shows a comparison of the results of the present study, two similar primate studies, and three non-primate studies that classify animal behavior using accelerometers and Random Forest models. Other studies using accelerometers may seek to distinguish between periods of activity and inactivity [30], overall activity patterns [68], or may be concerned with identifying just one type of behavior [69], requiring different methods and results metrics.

Boyd et al. (2004) [3] pose a definition of bio-logging as the "investigation of phenomena in or around free-ranging organisms that are beyond the boundary of our visibility or experience". Animals such as the Javan slow loris are small, arboreal, and nocturnal—all conditions that make them difficult for humans to observe in the wild. Bio-logging devices such as accelerometers effectively extend the capacity of our senses to allow us a previously inaccessible view into the activities and behaviors of animals such as the Javan slow loris, deep diving sunfish [21], flying and diving seabirds [43], or arctic muskox [70]. The information obtained from such studies is important to understand wildlife responses and resistance to global climate change, anthropogenic environmental modification and destruction [3,65,71]. At the same time, we can use bio-logging data to reconstruct environmental state and fluctuations since animal behavior is affected by the surrounding environment and therefore contains environmental information [9]. These insights can be integrated into ecosystem management programs to resist the effects of climate change and environmental degradation, including for species across a broad geographic range [72–74].

**Table 8.** Comparison of model accuracy and variable importance across a selection of studies which classified animal behaviour using accelerometers and Random Forest models.

| Author | Species | Number of Subjects | Accelerometer Model | Sampling Frequency (Hz) | Overall Accuracy | 3 Most Important Variables | | |
|---|---|---|---|---|---|---|---|---|
| Present study | Javan slow loris *(Nycticebus javanicus)* | 1 (wild) | Technosmart Axy 5S | 25 | 94.60% | Static_Lateral | Static_DorsoVentral | Z axis |
| Nekaris et al. [13] | Javan slow loris *(Nycticebus bengalensis)* | 1 (captive) | Technosmart Axy 5S | 26 | 80.7% | Static_Lateral | Static_Dorsoventral | *Y* axis |
| Fehlmann et al. [15] | Chacma baboon *(Papio ursinus)* | 9 (wild) | Daily Diary sensor | 40 | 88.3% | Static *X* axis | Pitch | PSD1Z |
| Tatler et al. [45] | Dingo *(Canis dingo)* | 3 (captive) | LISD2H | 1 | 87% | Z axis | S.D. X | Mean X |
| Kleanthouse et al. [75] | Hebridian sheep *(Ovis aries)* | 8 (captive) | MetamorionR | 12.5 | 99.43% | ------------- Not given ------------- | | |
| Jeantet et al. [76] | Hawksbill and Green turtles *(Eretmochelys imbricata and Chelonia mydas)* | 2 (captive) | Wilog Acquisition Control Unit | 50 | 86.96% | Diff_Deep | Statix *X* axis | Min_VEDBA |
| Jeantet et al. [76] | Loggerhead turtle *(Caretta caretta)* | 1 (captive) | Wilog Acquisition Control Unit | 51 | 79.49% | Diff_Deep | Pitch | Static *X* axis |

## 5. Conclusions

Accelerometry is an exciting technology that has the potential to significantly advance the field of behavioral ecology. Improvements to size, battery life, and performance of accelerometer devices are developing rapidly enabling researchers to collect animal data movements over longer periods of time and on a growing variety of species. This study shows that the combined use of accelerometers and Random Forest models can identify resting, locomotion, and feeding behaviors in a Javan slow loris. In order to validate this model, we needed to be able to observe a small nocturnal primate continuously for multiple nights, which was a great challenge. This limited the sample size to a single individual, and restricted the behavior we could observe during that period. For example, not enough social behaviors were seen for the analysis. With this validation, however, we now can apply the model to multiple individuals and begin to code behaviors in more detail. This methodology is beneficial to behavioral ecologists in cases where direct observation is limited and provides an alternative to habituation, which can be very challenging and time consuming. Information gained from biotelemetry and machine learning techniques can also be applied to conservation initiatives and can play a significant role in the protection of the world's biodiversity.

**Author Contributions:** Conceptualization, K.A.I.N., M.C. (Marco Campera), K.H. and A.H.; methodology, A.H., M.C. (Marco Campera), M.C. (Marianna Chimienti), K.A.I.N. and K.H. ; software, A.H., M.C. (Marco Campera) and M.C. (Marianna Chimienti); validation, A.H., M.C. (Marco Campera), K.H. and M.C. (Marianna Chimienti); formal analysis, A.H., M.C. (Marco Campera) and M.C. (Marianna Chimienti); investigation, K.A.I.N., K.H., E.A. and N.A.; resources, K.A.I.N.; data curation, K.A.I.N. and K.H.; writing—original draft preparation, A.H.; writing—review and editing, A.H., K.A.I.N. and M.C. (Marco Campera); visualization, A.H., M.C. (Marco Campera) and M.C. (Marianna Chimienti); supervision, K.A.I.N., M.A.I. and M.C. (Marco Campera); project administration, K.A.I.N., K.H. and M.A.I.; funding acquisition, K.A.I.N. All authors have read and agreed to the published version of the manuscript.

**Funding:** This study was funded by Augsburg Zoo, Cleveland Zoo and Zoo Society, Rewild's Primate Action Fund and Margot Marsh Biodiversity Fund, Disney Worldwide Conservation Fund, International Primate Protection League, Lee Richardson Zoo, Mohamed bin al Zayed Species Conservation Fund (152511813), Moody Gardens Zoo, NaturZoo Rhein, Omaha's Henry Doorly Zoo, People's Trust for Endangered Species, Plumploris E.V., San Francisco Zoo, Shaldon Wildlife Trust, Sophie Danforth Conservation Biology Fund and Zoo De Lille.

**Institutional Review Board Statement:** All research was approved by the Animal Care Subcommittee of Oxford Brookes University number OBURASC0911 and adhered to the ASAB/ABS Guidelines for the Use of Animals in Research. All necessary research permits were obtained from the Indonesian government. All research adhered to the legal and ethical guidelines of the Indonesian Institute of Sciences, Department of Wildlife and Department of Forestry.

**Informed Consent Statement:** Not applicable.

**Data Availability Statement:** The data presented in this study are available by request from the corresponding author.

**Conflicts of Interest:** The authors declare no conflict of interest. The funders had no role in the design of the study; in the collection, analyses, or interpretation of data; in the writing of the manuscript; or in the decision to publish the results.

## Appendix A

Following is the Random Forest script we used in RStudio to classify behaviors.
Acc_Data<-read.csv(file.choose(), header=TRUE) #open file containing the behaviors and acc data install.packages("randomForest")
library(randomForest)
set.seed(12345)
data_set_size <- floor(nrow(Acc_Data)*0.7)

```
indexes <- sample(1:nrow(Acc_Data), size = data_set_size)
training <- Acc_Data[indexes,] #corresponds to the 70%
validation1 <- Acc_Data[-indexes,] #corresponds to the 30%
training$Metadata = factor(training$Metadata)
rf_classifier = randomForest(Metadata ~ ., data=training, mtry=5, importance=TRUE) #this
runs the model
importance(rf_classifier)
varImpPlot(rf_classifier)
rf_classifier #accuracy for training (70%) dataset
        predValid <- predict(rf_classifier, validation1, type = "class") #this predicts the remain-
ing 30% of the daset based on random forest model
mean(predValid == validation1$Metadata) #mean prediction accuracy table(predValid,
validation1$Metadata)
        getTree(rf_classifier) MDSplot(rf_classifier, k=2) plot(rf_classifier)
        ACC_variables<-read.csv(file.choose(), header=TRUE) #this is to open the file with
importance variables. Make a 2-column dataset using the matrices from above and save as
separate file. ACC_variables$variable = with(ACC_variables, reorder(variable, importance,
median)) #reorders in order of importance
        install.packages("ggplot2")
library(ggplot2) #this summons the package that makes the box plot
install.packages("viridis")
library(viridis) png(filename="C:/Users/Location/Desktop/ImportanceAccelerometer
Movements.png", res=1000, units="in", width=9, height=5) #or tiff()
ggplot(ACC_variables, aes(x=variable, y=importance, fill=variable)) + geom_boxplot()+
theme(text = element_text(size=16))+ scale_fill_viridis(discrete = T, option="B", begin=0.95,
end=0.05)+ xlab("Variables in random forest") + ylab("Importance in random forest classi-
fier")+ theme(axis.text.x=element_blank(), axis.ticks.x=element_blank())+theme(panel.grid
= element_blank(),axis.line = element_line(colour = "black", size = 1, linetype = "solid"),
panel.background = element_blank())+theme(legend.title = element_blank())+scale_x_ dis-
crete(limits = rev(levels(ACC_variables$variable)))+guides(fill = guide_legend(reverse =
TRUE))
dev.off()
        png(filename="C:/Users/Location/Desktop/RFClassifierAccelerometerMovements.png",
res=1000, units="in", width=9, height=5)
varImpPlot(rf_classifier)
dev.off()
```

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
