# Peer review of "Analysis of Accelerometer Data Using Random Forest Models to Classify the Behavior of a Wild Nocturnal Primate: Javan Slow Loris (Nycticebus javanicus)"

_2673-4133, doi:10.3390/ecologies4040042_

Round 1

Reviewer 1 Report

This manuscript provides a proof of concept for the use of accelerometers in recording behaviors of slow loris.  It is a potentially useful method for species that are otherwise difficult to observe, but as the authors themselves note, a sample size of one means that the data presented here are not necessarily reliable.  Nonetheless, the authors have carefully elucidated

their methods so that others could replicate them, and the data they provided suggest which movement variables may be most useful and accurate, which is helpful to future researchers  on this and possibly other similar species.

Abstract

Please give common name of species­.

l. 19  ‘…species that are difficult to observe due to their cryptic nature of habitat inaccessibility.’ 

Not sure what this means – maybe this:  species that are difficult to observe due to their cryptic nature and/or to the inaccessibility of their habitat.’

l. 20  Explain briefly what a supervised approach is, given that this is a journal with a wide range of expertise in its readers.

l. 23  How is ‘resting’ a movement? 

Introduction

l. 35  What is a ‘behavior’ versus an ‘activity’ budget, which should be listing behaviors?  Please clarify.

l. 53  Capitalize ‘Pygosclis

l. 55 …species that were previously inaccessible.

l. 59. How is reflective eye shine a metric?  Please clarify.

Rewrite sentence that begins on l. 76 (‘For instance….’) for clarity.

Table 1 – organize by taxa rather than by date.  As it is, Aotus is in two places.  Also consider providing higher-level taxa to clarify relationships. 

l. 76:  Rewrite this sentence for clarity: ‘For instance, battery life can be affected by weather and humidity and may vary throughout the year according to seasons and is something to consider when planning deployment and retrieval of devices in the field [30,31,32].’   

l. 96 – 97  Reword:  ‘The data …are complex and often very large.’  Maybe you mean the entire data set that is generated in an observation is large?

Methods

It is somewhat problematic that olfaction is here considered to be associated only with foraging, given that mammals use olfaction in varied ways, including for example detection of predators and conspecifics, which they will do even during bouts of foraging.

Table 2:  This is difficult to read.  Perhaps add lines between entries, or put the description level with the behavior/abbreviation instead of vertically centered on it.  Also note that ‘abbreviation’ does not have enough space to stay on one line in the header.  The final column might also be easier to read if it were only left justified.

L. 193-196.  This sentence is confusing:  ‘We set the accelerometers to record at an interval of 25 Hz and can last up to 60 days at this rate, although battery life can be affected by temperature and humidity and a similar model has displayed reduced battery life in the tropics during months with the most rainfall.’

Something has been omitted after ’25 Hz and’ – possibly ‘the battery.’  Also,  the last parts of the sentence should be rewritten for readability.

204:  ‘from’ rather than ‘form’

202:  ‘We added behavioral data later to prepare it for the model’ is not clear.  Added the data to what?  Which behavioral data?

Table 3:  What is ‘Actiwatch?’  Is it a proprietary name for data recorded by the accelerometer? 

209:  typo – should be ‘from’ rather than ‘form’

215-216:  This is a run-on sentence.

243.  no capital letter for ‘Where’

Results

Table 5 caption – please re-word the first sentence for clarity.

L. 331-332.  The part after the semicolon is a sentence fragment.

Discussion

In l. 359, is the text still discussing the results of Nekaris et al., or has it returned to focus on this study presented here?

In several places in the MS, it is difficult to follow the writing.  Here is an example:

To do this, more calibrated direct behavioral observations must be taken on lorises wearing accelerometers.

How are direct behavioral observations supposed to be calibrated??  Maybe this means that humans should observe lorises wearing accelerometers to validate the findings presented here?

Author Response

ecologies-2582467

Analysis of accelerometer data using Random Forest models to classify the behavior of a wild nocturnal primate: Javan slow loris (Nycticebus javanicus)

We thank the editor and the reviewers for their very useful comments, which we can see improve the quality of the manuscript. Please see below our point-by-point response to these helpful reviews.

Response to reviewers

Reviewer 1

This manuscript provides a proof of concept for the use of accelerometers in recording behaviors of slow loris.  It is a potentially useful method for species that are otherwise difficult to observe, but as the authors themselves note, a sample size of one means that the data presented here are not necessarily reliable.  Nonetheless, the authors have carefully elucidated their methods so that others could replicate them, and the data they provided suggest which movement variables may be most useful and accurate, which is helpful to future researchers  on this and possibly other similar species.

We thank the reviewer for this point. Indeed, we are now employing accelerometers on many more individuals, but we did not want to proceed to that point until we had done model training with a closely related captive species (already published) and then validated with a wild one. The nature of the continuous sampling at night was very arduous and it took many nights to get the animal in view long enough. But with this validation, we can now apply it to the much larger sample size in future.

Abstract

Please give common name of species­.

Response: added

  1. 19  ‘…species that are difficult to observe due to their cryptic nature of habitat inaccessibility.’  

Not sure what this means – maybe this:  species that are difficult to observe due to their cryptic nature and/or to the inaccessibility of their habitat.’

Response: we agree – of was meant to be or ; we have changed the text to read  - or dense or difficult to access habitats

  1. 20  Explain briefly what a supervised approach is, given that this is a journal with a wide range of expertise in its readers.

Response: Thank you for pointing this out – we have added e.g., by observing in detail with a detailed ethogram the behavior of an individual wearing an accelerometer,

  1. 23  How is ‘resting’ a movement? 

We have deleted the phrase to clarify that this is a simple behaviour

Introduction

  1. 35  What is a ‘behavior’ versus an ‘activity’ budget, which should be listing behaviors?  Please clarify.

Deleted the phrase regarding activity budget as it was unnecessary

  1. 53  Capitalize ‘Pygosclis

Done

  1. 55 …species that were previously inaccessible.

Done

  1. 59. How is reflective eye shine a metric?  Please clarify.

Altered and added radio tracking

Rewrite sentence that begins on l. 76 (‘For instance….’) for clarity.

Altered and shortened

Table 1 – organize by taxa rather than by date.  As it is, Aotus is in two places.  Also consider providing higher-level taxa to clarify relationships.  

Decided to remove the table as it was a combination of diurnal and nocturnal taxa and did not make the appropriate point

  1. 76:  Rewrite this sentence for clarity: ‘For instance, battery life can be affected by weather and humidity and may vary throughout the year according to seasons and is something to consider when planning deployment and retrieval of devices in the field [30,31,32].’   

Altered and shortened

  1. 96 – 97  Reword:  ‘The data …are complex and often very large.’  Maybe you mean the entire data set that is generated in an observation is large?

Altered to show that that the detailed datasets require complex analyses

Methods

It is somewhat problematic that olfaction is here considered to be associated only with foraging, given that mammals use olfaction in varied ways, including for example detection of predators and conspecifics, which they will do even during bouts of foraging. 

The first author did the accelerometer validation analysis but not the field behavioural data collection. Indeed, for this study olfaction was not considered here. It only is in the sense that an exploring loris is more focussed on their environment, moving more slowly and perhaps sniffing, than a travelling loris who is clearly fast and direct. I have edited the table accordingly to clarify exactly what was meant for this validation.

Table 2:  This is difficult to read.  Perhaps add lines between entries, or put the description level with the behavior/abbreviation instead of vertically centered on it.  Also note that ‘abbreviation’ does not have enough space to stay on one line in the header.  The final column might also be easier to read if it were only left justified.

Have made it into three tables

  1. 193-196.  This sentence is confusing:  ‘We set the accelerometers to record at an interval of 25 Hz and can last up to 60 days at this rate, although battery life can be affected by temperature and humidity and a similar model has displayed reduced battery life in the tropics during months with the most rainfall.’ – edited to show that it actually did last this long as the manufacturer suggested

Something has been omitted after ’25 Hz and’ – possibly ‘the battery.’  Also,  the last parts of the sentence should be rewritten for readability. 

Edited

204:  ‘from’ rather than ‘form’

Edited

202:  ‘We added behavioral data later to prepare it for the model’ is not clear.  Added the data to what?  Which behavioral data?

Redundant to the paragraph below so we deleted it

Table 3:  What is ‘Actiwatch?’  Is it a proprietary name for data recorded by the accelerometer?  

This was a mistake from a cut and paste from an older paper using a different type of accelerometer – we apologise

209:  typo – should be ‘from’ rather than ‘form’

Could not find a second form in searching the document so I hope I got it!

215-216:  This is a run-on sentence.

Divided into several

  1. no capital letter for ‘Where’

Fixed

Results

Table 5 caption – please re-word the first sentence for clarity.

Deleted table and added to text

  1. 331-332.  The part after the semicolon is a sentence fragment.

Added a subject and a verb – these included

Discussion

In l. 359, is the text still discussing the results of Nekaris et al., or has it returned to focus on this study presented here?

Have clarified and added “in the current study”

In several places in the MS, it is difficult to follow the writing.  Here is an example:

To do this, more calibrated direct behavioral observations must be taken on lorises wearing accelerometers. 

edited

How are direct behavioral observations supposed to be calibrated??  Maybe this means that humans should observe lorises wearing accelerometers to validate the findings presented here?

edited

Reviewer 2

The ability of the RFM to predict animal behavior was studied. The bright side of the manuscript is to provide practical details for the related topic. However, some points are missing (mentioned below) in the manuscript. Therefore, I would like to make some suggestions to improve the quality of the manuscript as below: 

We thank the reviewer for their comments.

Abstract:

I think the abstract needs to be rephrased and improved. In my opinion, the authors need to supplement the importance of the results and how the results may be useful for further research. Authors may also say in 2-3 sentences that their findings contribute to behavioral ecology research and the conservation of species. In this way, the bridge between the problem and the solution found by the authors would be stronger. In my opinion, it is always good to finish the abstract with such a sentence. 

Have changed it to read: The methods used in this study can serve as guidelines for future research for slow lorises and other ecologically similar wild mammals. These results are encouraging and have important implications for understanding wildlife responses and resistance to global climate change, anthropogenic environmental modification and destruction, and other pressures.

Materials and methods:

I think it is necessary to add a picture of the study area and mark the land type.

We have one from a recent paper but it does not show rivers – Marco?

 Through the combination of graphics and text, it can help readers understand quickly and accurately. In addition, information on river hydrology, animal and plant resources in the region should be supplemented. In particular, the distribution of rivers and the types of animals and plants that may have an impact on the study of species behaviour. This paper is to investigate whether the RFM model can accurately predict the behavior of the species studied. So the various factors that can have an impact on the behavior of the species should be fully supplemented.

Line 153-158 Please rephrase here. 

Discussion:

The article does not seem to be comprehensive about the classification of behavior types.In addition to exploring, walking, and alerting, I think it's equally important to avoid harm and avoid carnivores.

The intention of this article is to provide the baseline validation, which is typical of accelerometer studies, so that we can indeed increase our sample size later once we know the model works. Of course there are behaviours that we could not yet capture which are of ecological importance such as active predation, or predator avoidance. This can be either addressed with increased sample size/observations in captivity or also accounting for the fact that some behaviours might never be observed in captivity or the wild. Indeed in 12 years, observing over 145 wild individuals, we only had evidence for predation three times. For the latter, unsupervised machine learning approaches or hidden Markov models can be used on data collected from wild species (Chimienti et al. 2016, 2021, 2022) and then compared to datasets and models obtained in captivity.  Moreover, combination of GPS-accelerometer metrics with biotic and abiotic factors will help to better understand why and how changes in behaviour occur. Validation of such behaviours, even partial, provides a robust estimate of how animals move.  This can be considered in later studies. See for example https://www.pnas.org/doi/full/10.1073/pnas.2004592118

In the relationship between intraspecific competition and interspecific predation. Its behavior maintains the stability of the ecosystem to a certain extent and promotes its operation at the same time. However, the author did not explain the relevant content. In addition, this paper does not seem to be closely related to Javan slow loris.

We could not focus on the relatively rare social behaviour because this study was part of a several step process. First, we had to gain data from captivity, where only a single animal was available, and thus by default we could not have a model for social behaviour. In addition, we have to be able to follow a wild individual continuously to further the validation process – it transpired that on those focal follow nights he was not engaged in many social behaviours. Thus we did not include those in this study as they were far too few for the models. Now that we have a validated model, we can continue to add new and rarer behaviours.

The content does not highlight the characteristics of group-living species. In particular, the author selected the primate as the object of study. But the article did not mention that the existence of certain specific behaviors in group-living species groups. Compared with other animals, their particularity and uniqueness need to be reflected and elaborated in the article.

We have added more behavioural detail about the loris in the “loris” paragraph second from last to introduction and described in the methods why we could not include social behaviours.

At present, there are still some deficiencies in the classification and identification of behavior types in this article. 

The slow loris is a nocturnal primate and a solitary forager. They are semi-gregarious, but it just so happens during the observations of the validation, no social behaviour took place. This needs to be considered in our later studies, now that the model is validated.

The author points out that this article is of great significance in the impact of global change on wildlife .There is a lot of literature on the impact of climate change on species. However, only 1-2 sentences are used to explain the relevant content in the article. It is suggested to supplement the relevant content and literature. I think the use of more specific content can help readers closely combine the results of the article with concepts such as global climate change. And then help to improve the overall quality of the discussion.

Line 394 Please rephrase here. 

Have added more references, details and rephrased

Conclusions:

The limitations of the study should be given in the conclusion section.

We have added three sentence regarding the limitations, especially regarding the social behaviour as stated above.

Reviewer 2 Report

The ability of the RFM to predict animal behavior was studied. The bright side of  the manuscript is to provide practical details for the related topic. However, some points are missing (mentioned below) in the manuscript. Therefore, I would like to make some suggestions to improve the quality of the manuscript as below: 

Abstract:

I think the abstract needs to be rephrased and improved. In my opinion, the authors need to supplement the importance of the results and how the results may be useful for further research. Authors may also say in 2-3 sentences that their findings contribute to behavioral ecology research and the conservation of species. In this way, the bridge between the problem and the solution found by the authors would be stronger. In my opinion, it is always good to finish the abstract with such a sentence. 

Materials and methods:

I think it is necessary to add a picture of the study area and mark the land type. Through the combination of graphics and text, it can help readers understand quickly and accurately. In addition, information on river hydrology, animal and plant resources in the region should be supplemented. In particular, the distribution of rivers and the types of animals and plants that may have an impact on the study of species behaviour. This paper is to investigate whether the RFM model can accurately predict the behavior of the species studied. So the various factors that can have an impact on the behavior of the species should be fully supplemented.

Line 153-158 Please rephrase here. 

Discussion:

The article does not seem to be comprehensive about the classification of behavior types.In addition to exploring, walking, and alerting, I think it's equally important to avoid harm and avoid carnivores.

In the relationship between intraspecific competition and interspecific predation. Its behavior maintains the stability of the ecosystem to a certain extent and promotes its operation at the same time. However, the author did not explain the relevant content. In addition, this paper does not seem to be closely related to Javan slow loris. The content does not highlight the characteristics of group-living species. In particular, the author selected the primate as the object of study. But the article did not mention that the existence of certain specific behaviors in group-living species groups. Compared with other animals, their particularity and uniqueness need to be reflected and elaborated in the article.

At present, there are still some deficiencies in the classification and identification of behavior types in this article. 

The author points out that this article is of great significance in the impact of global change on wildlife .There is a lot of literature on the impact of climate change on species. However, only 1-2 sentences are used to explain the relevant content in the article. It is suggested to supplement the relevant content and literature. I think the use of more specific content can help readers closely combine the results of the article with concepts such as global climate change. And then help to improve the overall quality of the discussion.

Line 394 Please rephrase here. 

Conclusions:

The limitations of the study should be given in the conclusion section.

Author Response

(The authors gave the same response as above.)
